# More Flexible Proximity Wildcards Path Planning with Compressed Path Databases

## Abstract

Grid-based path planning is one of the classical problems in AI, and a popular topic in application areas such as computer games and robotics. Compressed Path Databases (CPDs) are recognized as a state-of-the-art method for grid-based path planning. It is able to find an optimal path extremely fast without state-space search. In recent years, researchers tend to focus on improving CPDs from reducing CPD size or improving lookup performance. Among various methods, proximity wildcards is one of the most proven improvements in reducing the size of CPD. However, its proximity area is significantly restricted by complex terrain, which has more significant impacts on pathfinding efficiency and generates more additional costs. In this paper we enhance CPDs from the perspective of improving search efficiency and reducing search costs. Our work is to break the limitation between length and width of the proximity area, and adopt more flexible approaches to avoid obstacles, so as to reduce its impact on the proximity area and improve the search performance. Experiments performed on the benchmarks from Grid-Based Path Planning Competition (GPPC) demonstrate that the two proposed methods can effectively improve search efficiency and reduce the search costs by 2-3 orders of magnitude. Remarkably, our methods can further reduce storage costs, and improve compression capability of CPDs simultaneously.

## Introduction

Path planning is one of the important problems in artificial intelligence which has been studied for many years and widely applied in real scenes such as robotics and computer games (Freund and Hoyer 1986; Cui and Shi 2011). Grid-based path planning is one of the most active research directions on this problem. Over the years, numerous excellent algorithms have been proposed, and the Grid-Based Path Planning Competition (GPPC) (Sturtevant et al. 2015) is the cornerstone in the development of these novel path-planning achievements. The impact of these advancements is essential and far-reaching, and they are poised to revolutionize path planning in the coming years (Botea 2011; Uras and Koenig 2014; Harabor and Grastien 2014; Rabin and Sturtevant 2016; Sturtevant and Rabin 2016; Uras and Koenig 2017; Goldenberg et al. 2017; Cohen et al. 2017; Harabor and Stuckey 2018; Salvetti et al. 2018; Hu et al. 2019, 2021).

The Compressed Path Databases (CPDs) (Botea 2011; Botea and Harabor 2013) known as a group of state-of-the-art techniques for grid-based path planning, is designed to speed up the response and reduce the first move delay during the agent's path planning tasks (Zhao 2022). Each CPD is a data structure that provides the optimal first move from any start cell $s$ to any target cell $t$. The process of finding an optimal path is to iteratively search such a group of CPDs to find the optimal move directions to reach the target without any state space search. The main disadvantage of CPDs with ultra-fast pathfinding speed is the huge build cost. Each CPD needs to be encoded by all-pairs of pre-computation, and then compressed with encodes and store the result. In recent years, researchers mainly tend to focus on reducing the size of the CPD such as single row compression (SRC) (Strasser, Harabor, and Botea 2014), heuristic redundant symbols (Chiari et al. 2019), proximity wildcards (Chiari et al. 2019), bidirectional wildcards (Salvetti et al. 2017), or improving lookup performance such as two-oracle path planning algorithm (Topping) (Salvetti et al. 2018). Among them, the SRC performed best in GPPC 2014, and has become an important baseline for improving CPDs.

The CPDs improved by proximity wildcards (CPDs_PW), replaces the storage symbols of qualified nodes in the largest square centered on any node with wildcards to reduce the preprocessing memory on the basis of heuristic redundant symbols. Their experiments are mainly based on Dragon Age: Origins (DAO) (Sturtevant 2012), which verifies the excellent compression capability of CPDs_PW. It can achieve better on 99% of the maps, and can even compress the map size to 1/55 of SRC on the map AR0044SR.

Despite the effectiveness of CPDs_PW for compression is already advanced, there are still the following drawbacks:

1. Severely limited by complex terrain. Its area of concern is square in shape, which causes it to miss out on opportunities for expansion and incur more search costs when it encounters obstacles in any direction.

2. Inefficient search. The compression is not efficient enough, resulting in more binary searches to find the target from during pathfinding, leading to inefficient search.

In this paper, our idea for the above problems is to expand the proximity area by breaking its shape limitation without weakening the compression capability. We propose the following two methods:

1. Rectangular Proximity Wildcards (RPW): This method breaks the limitation of length and width, and expands the shape of proximity area to a rectangle, so that when obstacles are encountered in any direction of length or width, the other that does not encounter obstacles can continue to be expanded.

2. Coordinates Proximity Wildcards (CPW): This method breaks the limitations of traditional geometry. In order to avoid obstacles more flexibly and improve the expansion opportunities, four quadrants are divided with the current node as the center, and the largest rectangle is expanded in each quadrant. The areas drawn by the four quadrants are all proximity areas of the current node.

We also expand the scale of the experiment, adding Baldurs Gate II (BGII) (Sturtevant 2012) and Starcraft (Sturtevant 2012) on the top of DAO, a total of 351 maps, and take SRC and CPDs_PW as the experimental baselines. The experimental results show that the two methods we proposed can effectively improve the search efficiency and reduce search costs by 2-3 orders of magnitude. In addition, they can also improve compression capability of CPDs. Experiments demonstrate that the more flexible the method is to avoid obstacles, the less it is affected by complex terrain, and the more obvious the improvement in efficiency.

The paper is structured as follows: First, we provide an overview of related work. Then, we introduce the fundamental principles of the CPDs and the key technologies utilized. After that, we present a detailed description of the RPW and CPW, along with description diagrams and algorithm pseudo codes. We also present the experimental results and analysis, and conclude the paper at the end.

## Related Work

As a leading technique for optimal pathfinding, CPDs (Botea 2011) has received extensive attention since it was proposed. Its main idea is to achieve speedup through preprocessing and additional memory, which has shown excellent performance in solving speed, but its huge memory requirements seem to be a huge burden.

Therefore, exploring better compression methods for CPDs has become a research hotspot. The Copa (Botea and Harabor 2013), which combined list trimming, run length encoding, and sliding window compression to improve compression capabilities at the cost of lost time, is one of the best competitors in GPPC 2012. In 2014, SRC (Strasser, Harabor, and Botea 2014), an algorithm combined with run length encoding (RLE) to compress rows, outperforms Copa in both compression and query time, and became one of the winners in GPPC 2014. Although SRC has excellent performance in compression ability and speed, it still requires huge memory overhead on large maps. In order to solve this practical bottleneck, wildcard substitutions (Salvetti et al. 2017) are introduced, which uses wildcards to replace part of the CPD-encoded data, and can be combined with heuristics to reduce the size of CPD. Its main idea is that given any start node $s$ and target $t$, as long as either route from $s$ to $t$ or $t$ to $s$ is feasible, a complete optimal path can be established. This method is proven to be effective, and the proximity wild-

cards (Chiari et al. 2019) is its extension with a new concept of redundant symbols proposed at the same time. The proximity wildcards uses redundant symbols to define a square centered on the current node, which is called the proximity area, and all nodes in it can be optimally reached through the heuristic move. Experiments verified that proximity wildcards is one of the most effective methods to improve the compression capability of CPDs, but it is seriously affected by obstacles and therefore runs with limited efficiency. Our work is related to proximity wildcards, but we choose to reduce the impact of obstacles on efficiency in more flexible ways (e.g. rectangle, quadrants).

The above works are all approaches to reduce the memory overhead of CPDs with the goal of finding optimal solutions, often accompanied by a significant loss of preprocessing time. In 2020, a centroid-based bounded suboptimal method (Zhao et al. 2020) reduces storage costs by only calculating the first-move data of selected nodes, which ensures that path costs within the fixed bound of the optimal solution. This approach innovatively reduces preprocessing time and storage costs by trading some suboptimality.

There are also works focus on improving search performance, such as Topping (Salvetti et al. 2018), which is a combination of SRC and Jump Point Search+ (JPS+) (Harabor and Grastien 2014). It first calls SRC to find the best move direction from the current node to the target, and then calls JPS+ to judge the feasible steps in this direction. In most cases, Topping can be more than an order of magnitude faster than SRC, but at the cost of nearly doubling memory consumption. To deal with the huge memory overhead, TOPS (Hu et al. 2021) and Topping+ (Hu et al. 2021) are proposed. The TOPS reduces preprocessing memory by calculating only the CPD data of the jump points. The Topping+ extracts a series of complete paths from the successor start nodes to the target. Their search performance are competitive with Topping, while having smaller space requirements and better time performance.

## Background

**Gridmap.** A gridmap is a two-dimensional mobile agent operating environment, where each cell is either traversable (white squares) or obstacle (black squares). It has two common ways for defining neighboring relationships: 4-connected grids and 8-connected grids. In 4-connected grids, a traversable cell has at most four neighbors, and the movable directions are: East, South, West and North (symbolized as E, S, W, N), with a move cost of 1 per cell. As shown in Figure 1, for any node $n$ on the map, the 8-connected grid map allows diagonal movement, adding movable directions: Northwest, Northeast, Southwest, and Southeast (symbolized as NW, NE, SW, SE), the cost of linear movement $\vec{c}$ is 1, and the cost of diagonal movement $\vec{d}$ is $\sqrt{2}$ . A path planning problem on a gridmap starts at cell $s$ called start node and ends with cell $t$ called the target node. Its goal is to find the most efficient path with the minimum cost from the start node to the target node. The formula $s' = s + k\vec{d} + m\vec{c}$ ($k$ and $m$ are integers) indicates that moving from node $s$ to node $s'$ needs $k$ times along the

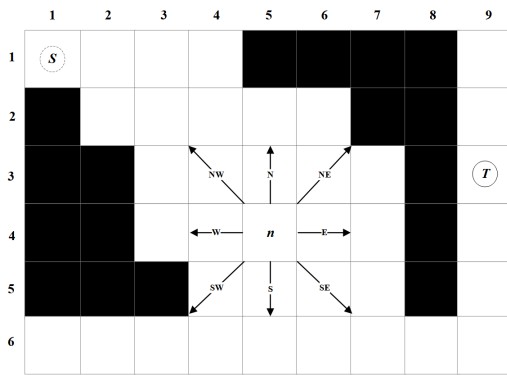

Figure 1: 8-Connected grid map.

| N | N | N | ■ | N | N,NE | N,NE | N,NE | N,NE |
|---|---|---|---|---|---|---|---|---|
| N,W | N | N | N | N | E,NE | ■ | E,NE | E,NE |
| W | ■ | | N | E,NE | | E,NE, SE | E,SE |
| W | W | W | W | s | E | E | ■ | E,SE |
| W | ■ | | S | SE | E,SE | E,SE | E,SE |

Figure 2: Example of the first move array for node $s$.

direction $\vec{d}$ and $m$ times along the direction $\vec{c}$. If $k$ and $m$ are negative integers, it means reverse movement. In this paper, we assume that diagonal movements are not allowed if they will touch an obstacle cell.

**Compressed Path Databases (CPDs).** CPD is a data structure that encodes the optimal first move from any node $s$ to a target node $t$ on a map. It is the result of compressing the first move array using run length encoding (RLE). (Strasser, Harabor, and Botea 2014). The first move array $T(s)$ of node $s$ stores the first move of the shortest path from node $s$ to each reachable node $t$, which is calculated by Dijkstra. CPDs are constructed by computing a first move array for each node in the map in offline preprocessing. Chiari et al. redefined CPD as consisting of the first move array (the original meaning of CPD) and auxiliary search data because of the particularity of their methods (such as proximity wildcards). We use this definition. According to Chiari et al.'s definition, the CPD size is the sum of the first move array size and the auxiliary data size. The size of the first move array represents the method's capability in compression. Each binary search extracts the next move in the first move array of the current node based on the target position.

**Run Length Encoding (RLE).** RLE compresses the first-move array by more compactly representing the substrings (called runs) composed of the repetition of the same symbol. It labels obstacles and source node with the wildcard "*", because they don't need to be searched. Taking Figure 2 as an example, the first row can be compressed into NNN*NNNNN. Usually we use a substring to represent the whole map, so the first row is represented as 1N, and this map is represented as 1N 15E 19W 23N 24E 28W 33E 37W 41S 42SE. Where the letters denote optimal first moves for moving the source node $s$ to the corresponding position and the numbers are subscripts starting from 1 after reducing the 2D first-move array to a 1D array.

**Heuristic move.** Heuristic move refers to the first move of node $s$ to ignore obstacles and move towards the goal. It is determined by the shortest heuristic distance calculated by heuristic function $F_e(n,t)$, where $\omega(s,n)$ is the cost from source $s$ to node $n$ and $f_e(n,t)$ is the heuristic distance function from $n$ to $t$. In this paper we use the Euclidean

distance to calculate the heuristic move. We also use redundant symbol $h$ to mark the nodes where the heuristic move coincides with the first move to improve efficiency. That is, if the $F_e(n,t)$ returns a first move belonging to the first move array $T(s)$, add the redundant symbol $h$ to $T(s)$.

$$f_e(s,t) = \sqrt{(s.x - t.x)^2 + (s.y - t.y)^2} \quad (1)$$

$$F_e(s,t) = \arg\max_{(s,n) \in E}\{\omega(s,n) + f_e(n,t)\} \quad (2)$$

**Proximity Wildcards**. Proximity wildcards computes the largest square proximity area centered on $s$, where traversable nodes must have the heuristic redundant symbol $h$. If the target node $t$ is within the proximity area of $s$, the heuristic move can be applied directly, otherwise look for the first move.

## Rectangular Proximity Wildcards

Typically, when using the proximity wildcards, nodes within the proximity area of node $s$ can usually use heuristic moves as the first move to optimally reach. However, its proximity area is square, meaning that obstacles in either direction will cause the proximity area to stop expanding, so this method is not effective in complex terrain. In this section, we introduce a method called Rectangular Proximity Wildcard (RPW), whose region of interest is rectangular, its outstanding advantage is that when obstacles are encountered in either direction of length and width, the other direction is unaffected and can still continue to be expanded, as shown in Figure 3.

**Definition 1**: Given a node $s$ and a function $F_x(s,t)$, the width of largest proximity rectangle $R$ centered on $s$ is expressed as $rec(s).x$ and length is expressed as $rec(s).y$, for any node $n \in R$, there is $f_x(s,n) \in T(n)$, $T$ is the first-move array of $s$, and $rec(s).x * rec(s).y$ is the maximum value $d \in N$.

The cells in the proximity rectangle are all optimally reachable from $s$ by heuristic moves, and we denote each such cell with the wildcard character "*". As depicted in Figure 3, the compression result of the first move array with the proximity wildcards is 1N; 5h; 10N; 14h; 19W; 26h; 27E; 28h; 37W; 43h, a total of 10 RLE runs. On the other hand, the compression result of the first move array with RPW is 1N; 5h; 10N; 14h; 19W; 27E; 28h; 37W; 45h, a total of 9 RLE runs.

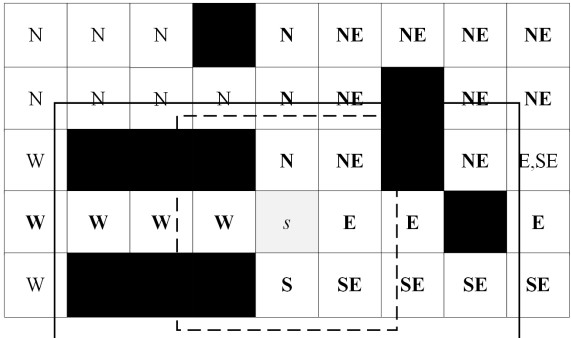

Figure 3: Proximity areas. RPW's line is solid while proximity wildcards 's line is dashed. Source node is shown in gray. Heuristic moves coinciding with the first moves are shown in bold.

---

**Algorithm 1: CPDHRP$(s, t)$**

**Input**: start node $s$, target node $t$
**Output**: first-move $m$

1: $X \leftarrow rec(s).x.$
2: $Y \leftarrow rec(s).y.$
3: **if** $|s.x - t.x| \leq \frac{X}{2} \wedge |s.y - t.y| \leq \frac{Y}{2}$ **then**
4:     **return** $F_x(s, t)$
5: **else**
6:     $m \leftarrow CPD(s, t)$
7:     **if** $m = h$ **then**
8:         **return** $F_x(s, t)$
9:     **else**
10:        **return** $m$
11:    **end if**
12: **end if**

---

RPW has a larger proximity area than proximity wildcards, which typically leads to more efficient compression and more readable encoding due to the need for fewer RLE runs. This solid foundation for the improvement of search performance. In the online search, RPW is able to flexibly respond to complex terrain changes, and its larger proximity area can help to use more heuristic moves, which effectively improves search efficiency and reduces cost. Algorithm 1 presents the CPD search function CPDHRP$(s, t)$ that utilizes the RPW and Algorithm 2 shows the improved CPDs algorithm CPDs_RPW$(s, t)$ with the CPDHRP$(s, t)$.

## Coordinates Proximity Wildcards

In the previous section, we introduced a flexible method to improve search performance, and it also has a certain effect on compression. Although RPW has effectively improved the search efficiency of CPDs, the use of traditional geometry as proximity areas still has significant limitations. When confronted with extremely complex and narrow maps (such as bridge maps), in more cases, the area of RPW would be very restricted, even as large as proximity wildcards. In this section, we propose a more flexible method called Coordinates Proximity Wildcards (CPW), as shown in Figure 4.

---

**Algorithm 2: CPDs_RPW$(s, t)$**

**Input**: start node $s$, target node $t$
**Output**: Path $p$

1: $p[].$
2: **while** $s \neq t$ **do**
3:     $(s, n) \leftarrow CPDHRP(s, t)$
4:     $p \leftarrow p + [(s, n)]$
5:     $s \leftarrow n$
6: **end while**
7: **return** $p$

---

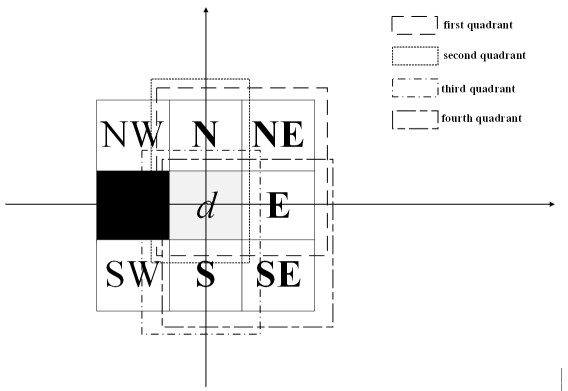

Figure 4: Proximity areas of CPW

**Definition 2**: Given a node $s$ and a function $F_x(s, n)$, the CPW area $C$ consists of four largest rectangles $R_m$ with $s$ as the corner. The length and width of each quadrant rectangle $R_m$ are expressed as $rec(s)_m.x$ and $rec(s)_m.y$, $m \in [1, 4]$. For any node $n \in C$, $f_x(s, n) \in T(n)$, $T$ is the first-move array of $s$.

We replace the traditional geometry centered on the source node with quadrants centered on the source node, and expand the largest rectangle in each quadrant. For the sake of illustration, we use PW here to represent the proximity wildcard method. As shown in Figure 5, a more flexible expansion method can make the proximity area larger. The area of the proximity area of CPW is three times larger than RPW, and six times larger than PW, which means more efficient search and compression. However, this method needs to store four times the size of auxiliary data, because each

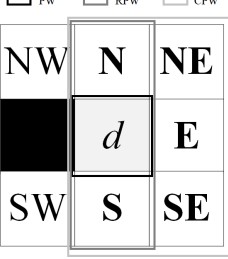

Figure 5: Comparison of proximity areas

of its proximity areas needs to store the side lengths of the four largest rectangles. Algorithm 3 checks if target node $t$ is within proximity area to any source node $s$. The CPD search function in Algorithm 4 utilizes CPW for device control, and is implemented in Algorithm 5.

---

**Algorithm 3:** GetCPW$(s, t)$

---

**Input**: start node $s$, target node $t$
**Output**: $true$ or $false$

1: **for** $n = 1$ to $4$ **do**
2:     $X.n \leftarrow rec(s)_n.x$.
3:     $Y.n \leftarrow rec(s)_n.y$.
4:     **if** $|s.x - t.x| \leq X.n \wedge |s.y - t.y| \leq Y.n$ **then**
5:         **return** $true$.
6:     **end if**
7: **end for**
8: **return** $false$.

---

**Algorithm 4:** CPDHCP$(s, t)$

---

**Input**: node $s$, node $t$
**Output**: first-move $m$

1: **if** GetCPW$(s, t)$ **then**
2:     **return** $F_x(s, t)$
3: **else**
4:     $m \leftarrow$ CPD$(s, t)$
5:     **if** $m = h$ **then**
6:         **return** $F_x(s, t)$
7:     **else**
8:         **return** $m$
9:     **end if**
10: **end if**

---

**Algorithm 5:** CPDs_CPW$(s, t)$

---

**Input**: node $s$, node $t$
**Output**: Path $p$

1: $p[]$.
2: **while** $s \neq t$ **do**
3:     $(s, n) \leftarrow$CPDHCP$(s, t)$
4:     $p \leftarrow p + [(s, n)]$
5:     $s \leftarrow n$
6: **end while**
7: **return** $p$

---

## Experiments

We performed experiments on three game benchmarks from GPPC (Sturtevant et al. 2015): BGII, DAO and Starcraft. The BGII has a total of 120 maps, mainly composed of small maps, with nodes distributed between 100 and 60,000. The DAO is mainly composed of medium-sized maps, and the number of nodes in its 156 maps is between 100 and 140,000. The Starcraft is a large map set with 75 members, ranging from 50,000 to 760,000 nodes. We use SRC and CPDs_PW as experimental baselines, the former is our main

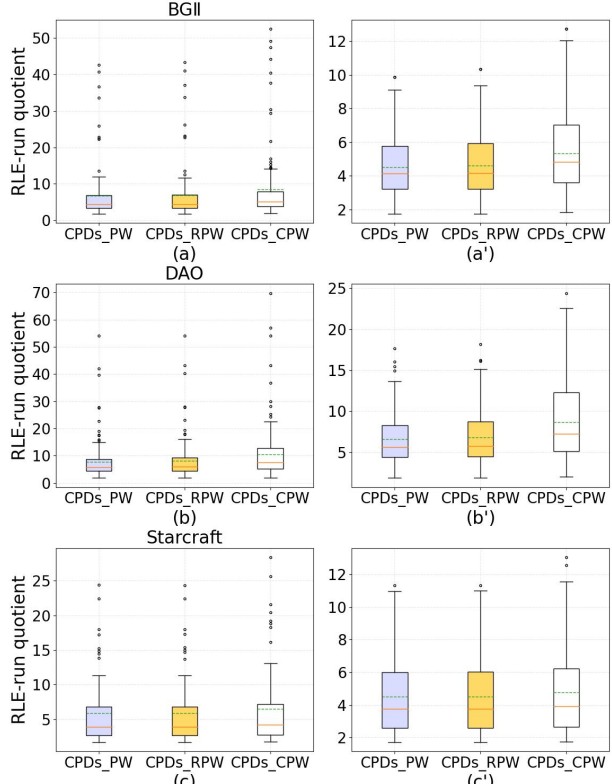

Figure 6: Compression cost factor $C_{RLE-runs}$. (x') is an enlarged version of (x)'s box. The compression cost factor is $C_{RLE-runs}$ for completing a compression task.

improvement and the latter is essential reference. After applying our methods to SRC, the resulting algorithms are as follows:

- CPDs_RPW, improved by rectangular proximity wildcards (RPW).

- CPDs_CPW, implements the coordinates proximity wildcards (CPW).

The experiments use runtime to measure search efficiency and the number of binary searches used to find the target during the search as the search costs. They are the two main metrics of the experiments. In addition, we also verify the compression costs through RLE runs and identify the compression capability with the first move array size. It is important to mention that the CPD for CPDs_PW, CPDs_RPW, and CPDs_CPW are all composed of the first move array and the auxiliary data, where the first move array size can illustrate the strength of compression capability, and the auxiliary data stores the proximity areas' sides. Due to the large difference in map size, there is a huge difference in data distribution. Therefore, for other performance metrics except runtime, we use the "factor" to describe the algorithms' performance. The factor is the result of dividing the SRC running results for the same map by the running results of other CPDs variants. Our definition of "factor" is shown in Equa-

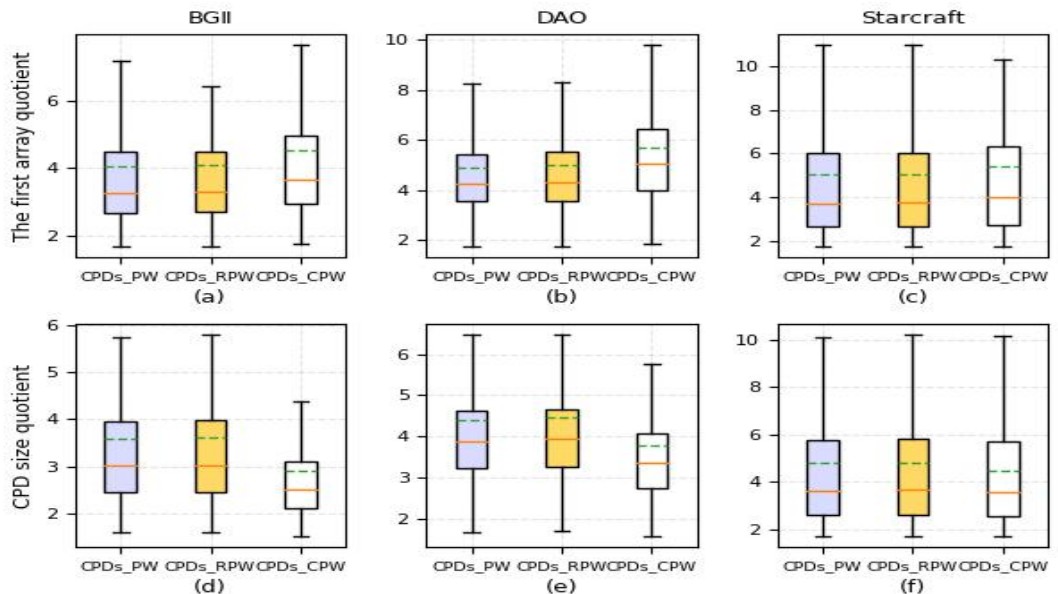

Figure 7: Compression capability factor $C_{first-move}$ and Memory factor $C_{CPD}$. (a) to (c) show the compression capability factor $C_{first-move}$, indicating the compression capability of methods. (d) to (f) show the memory factors $C_{CPD}$, indicating the memory occupied by the CPD size, and the CPD size is the sum of the first move array and auxiliary data sizes.

tion 3.

$$C_{metric_i} = \frac{realnum_{SRC}(metric_i)}{realnum_x(metric_i)} \quad (3)$$

$metric_i$ represents the $i$-th metric, and $realnum_x(metric_i)$ represents the real value of the $i$-th metric of algorithm $x$. $x$ can be assigned to CPDs_PW, CPDs_RPW, and CPDs_CPW. All algorithms are implemented in C++, and the experimental environment is Ubuntu 20.04.3 LTS, with processor AMD® Ryzen 9 5900*12core processer*24 and 31.4GiB RAM.

### Preprocessing

Before searching online, each map needs to be preprocessed offline. The use of more flexible proximity wildcards in the preprocessing stage reduces compression costs and improves compression capability.

The compression cost factor $C_{RLE-runs}$ is the quotient of RLE runs required by SRC and CPDs_X to complete a compression task. Higher compression cost factor means fewer RLE runs, leading to lower compression costs. As shown in Figure 6, our methods are effective in saving the compression cost, where CPDs_CPW has the highest $C_{RLE-runs}$ and the best performance. This means they can yield leaner compression results and lower compression costs, and contribute to search efficiency due to more readable compressed results.

Figure 7 (a), (b), (c) displays the distribution of compression capabilit $C_{first-move}$. A higher $C_{first-move}$ implies a smaller first move array size, which results in better compression performance. We can visualize that both of our methods have improved compression capability, with

CPDs_CPW performing best. From Figure 7 (e), (d), (f), it can be seen that the memory factor $C_{CPD}$ of CPDs_RPW is slightly higher than that of CPDs_PW, while CPDs_CPW has been at a disadvantage. It is due to the fact that the size of CPDs_CPW's auxiliary data (proximity distance) is four times larger compared to CPDs_PW and CPDs_RPW, resulting in larger CPD sizes and more memory costs. Therefore, sometimes we may encounter a scenario where the memory gain is lower than the expenditure. We can also see that as the map sizes increase, the memory factor of CPDs_CPW becomes higher and higher. It fully demonstrates the performance advantage of our methods in solving large-scale maps.

### Online Search

The search efficiency increases with shorter runtime, as shown in Figure 8. The runtime of our methods is significantly less than that of CPDs_PW. Because they have larger proximity areas which effectively reduces binary searches and makes the search faster. We can also find that CPDs_CPW takes more runtime than CPDs_RPW. Since CPDs_CPW's CPD lookup function is more complex, so each call takes more time. Both CPDs_RPW and CPDs_CPW's lookup functions are more complex than CPDs_PW's, so each call to the find function takes more runtime, but both take less time than CPDs_PW due to their huge search efficiency gains.

The binary search factor $C_{binary-search}$ is the quotient of the binary searches required by SRC and each CPDs variant when performing the same search task. If a CPDs variant performs a search that requires fewer binary searches, it

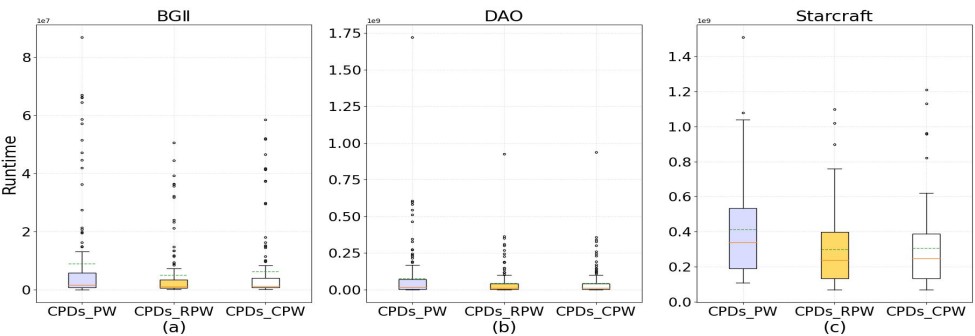

Figure 8: Runtime (unit: ns). Runtime is the time taken to perform an online search task, and we use it to measure search efficiency

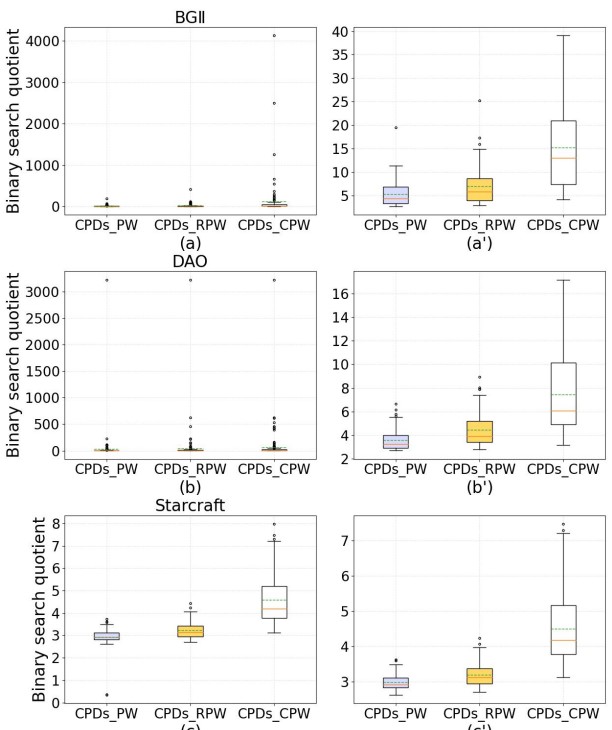

Figure 9: Binary search factor $C_{binary-search}$. (x') is an enlarged version of (x)'s box. The binary search factor $C_{binary-search}$ is the quotient of binary searches required by SRC and CPDs_X to complete a search task.

| Map | #Cell | The number of binary searches | | | |
|---|---|---|---|---|---|
| | | SRC | PW | RPW | CPW |
| AR0041SR | 2282 | 8258 | 131 | 76 | 2 |
| AR0418SR | 1428 | 2501 | 13 | 6 | 0 |
| AR0408SR | 354 | 1092 | 25 | 13 | 3 |
| orz105d | 679 | 2642 | 36 | 12 | 5 |
| orz107d | 637 | 1791 | 54 | 22 | 4 |
| lgr605d | 2983 | 8169 | 83 | 36 | 21 |

Table 1: Examples of exceptional binary searches. #Cell is the number of nodes contained in the map.

| Set | Number of wins | | | |
|---|---|---|---|---|
| | SRC | PW | RPW | CPW |
| Starcraft (75) | 1 | 15 | 19 | **28** |
| DAO (156) | 1 | 53 | 60 | **106** |
| BGII (120) | 0 | 92 | 102 | **115** |

Table 2: Comparative results with Topping on the number of binary searches. The number in brackets is the map number.

is equivalent to a higher binary search factor, implying less search cost. As shown in Figure 9, we can intuitively see the improvement effect of the three CPDs variants on the binary search, in which the $C_{binary-search}$ of CPDs_RPW and CPDs_CPW are remarkably higher than CPDs_PW. In both BGII and DAO, we experience remarkable effects that are magnified by hundreds or even thousands of times, despite taking place on small maps. Some of these exceptional cases are listed in Table 1. We find $C_{binary-search}$ stable and concentrated as map size increases. For example, CPDs_CPW has 14 (21-7) quartile distance in small map set BGII, 5 (10-5) in medium map set DAO, and less than 2 in large map set Starcraft. It is also evident that CPDs_CPW performs better than CPDs_RPW and CPDs_RPW outperforms CPDs_PW based on their median, mean, and quartiles. The average binary search factor $C_{binary-search}$ for CPDs_PW on BGII is 5.3, while CPDs_RPW's is 7.0 and CPDs_CPW's is 15.2, which is twice that of CPDs_RPW and three times that of CPDs_PW. On DAO maps, CPDs_CPW has an average binary search factor of 7.5, which is twice as much as

that of CPDs_PW and 1.7 times that of CPDs_RPW. Even on Starcraft, the average binary search factor of CPDs_CPW can reach 1.6 times that of CPDs_PW and 1.4 times that of CPDs_RPW.

It is because a more flexible proximity wildcards method avoids more obstacles, further reduces the impact of complex terrain, and allows for a larger proximity area. In other words, CPDs_CPW has the largest proximity area, while the area of CPDs_RPW is at least as large as that of CPDs_PW. A larger proximity area can enhance search performance from two different perspectives:

1. More powerful heuristics. Both theory and intuition indicate a larger proximity area means that more heuristic information can be used to find the target during the search process.

2. More effective search preparation. A larger proximity area in the preprocessing stage makes compression more efficient, resulting in more compact RLE encoding and efficient search.

### Our methods vs Topping

Our methods have enhanced the search performance of CPDs based on reducing the size of CPD. We compare SRC, CPDs_PW, CPDs_RPW, and CPDs_CPW, in terms of both search and compression. We also conduct comparative experiments with Topping, a representative algorithm in another research area, which significantly improves search performance but at the expense of huge memory and inefficient compression. Experiments verify the effectiveness of our methods, and CPDs_CPW even is competitive with Topping in search costs. The pathfinding process of Topping is jointly conducted by the SRC oracle and the JPS+ oracle. What we compare is only the number of binary searches of the SRC oracle in Topping. Experiment result show that CPDs_CPW performs best among variants, and has an absolute advantage in binary searches on 68% medium-sized maps and 96% small maps, and occupies a place in large maps. Noted that our methods utilize only half the storage cost used by Topping.

## Conclusion and Future Work

As a leading technology for path planning, CPDs has the bottleneck of huge storage overhead. A method called proximity wildcards significantly enhances the compression capability of CPDs with surprising results in reducing storage costs. However, it is severely limited by complex terrain, making the search inefficient and and generating more costs.

In this paper, we extend proximity wildcards and propose two methods RPW and CPW that can be more flexible in avoiding obstacles to compute larger proximity areas in response to complex terrain changes. Meanwhile, we extended the experiment scale to three full datasets. The experimental results demonstrate that rectangular proximity wildcards (RPW) and coordinates proximity wildcards (CPW) can flexibly deal with complex terrain, and significantly improve the search performance, and further improve its compression performance.

Our future work mainly focus on the following directions:

1. Exploring ways to reduce time while maintaining optimality, compression and search capability. Investigating new encoding alternatives to RLE is an interesting research direction.

2. Exploring a combination algorithm compatible with CPDs that has powerful search performance without causing internal interaction storage costs.

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
