# OpenReview forum: "More Flexible Proximity Wildcards Path Planning with Compressed Path Databases"
_icaps-conference.org/ICAPS/2024/Conference — ICAPS 2024_

### Official Review · Reviewer_K9zB · 2023-12-22

**Significance And Importance:** 2
**Soundness:** 4
**Novelty:** 3
**Clarity:** 2
**Overall Evaluation:** 1
**Confidence:** 5

**Weaknesses:**

0: Minor weaknesses requiring some work to be addressed for the paper to be accepted.

**Contributions Of The Paper:**

The paper proposes two flexible proximity wildcard methods to improve the compression of CPDs.

**Ethical Considerations:**

(1) Not Applicable: The paper does not have any ethical considerations to address

**Nomination For Best Paper:**

No

**Questions For Authors:**

What's the total query processing time of CPD_PW, CPD_RPW and CPD_CPW?

**Reproducibility:**

3: Authors describe the implementation and domains in sufficient detail.

**Strengths Of The Paper:**

Flexible proximity wildcards (RPW and CPW) can significantly increase the number of entries that are encoded with a wildcard,
thus reducing the number of binary searches and consequently improving query processing time.

**Weaknesses Of The Paper:**

The weaknesses of this work can be viewed from two aspects: method and presentation.

From a methodological perspective, while RPW and CPW improve the compression of CPDs, they require significantly more auxiliary data
to store wildcard information. This results in no significant reduction in CPD size.

In terms of presentation, in the experiment section, the author highlighted the reduction in the number of binary searches. However, a
decrease in binary searches doesn't necessarily equate to an improvement in processing time. Figure 8 presents the runtime
distribution, but the specific improvements are not clear.

It should also be noted that CPDs are already an "ultra-fast" path planning method—improving the processing time by several
percentages doesn't make a strong impact. What the authors should demonstrate in their experiment would be concepts like a "smaller
CPD size with no worse processing time" or a "significant CPD size reduction with slightly worse processing time"."

What's more, there are several claims that seem overstated or potentially misleading:

 1 In the 'preprocessing' section, the text states, 'It fully demonstrates the performance advantage of our methods in solving
   large-scale maps.' This is unclear, as only three map sets are referenced. There is no apparent trend correlating with map size.
 2 In the 'online search' section, it's stated that 'The runtime of our method is significantly less than that of CPDs_PW.' It can be
   argued that the runtime improvement seen in Figure 8 isn't notable enough to be classed as 'significant.'
 3 In the 'abstract', there is a claim that '...the two proposed methods can effectively improve search efficiency and reduce the
   search costs by 2-3 orders of magnitude.' Firstly, it's debatable whether the improvement actually reaches 2-3 orders of magnitude.
   Secondly, the number of binary searches is not synonymous with 'search cost.'

In summary, I perceive this work as an incremental work that has a limited impact.
---
Revision:
After reading the rebuttal and revisiting the paper, I finally realized that the major point the authors attempt to emphasize is
'a method that improves query processing without hurting memory consumption (and sometimes even benefits from it)'.
Thus I decide to remove 'method' from the weaknesses.

The writing still needs additional work to improve, more specifically:

1. Highlight the contribution. While the abstract does mention it, the emphasis seems to fade as one reads through the paper. The
   message that comes across is more 'we developed a novel wildcard method to reduce memory...' than the intended one.

2. Author should consider to change the order of experiment section - e.g., move the size comparison after speed/search comparison.

3. Add the result on elapsed time in experiment and discuss the impact of reducing number of binary search. E.g., concept like "cheaper CPUs get more benefit from less binary search..." can be useful.

In conclusion, these weaknesses can be fixed with some effort, therefore I change my decision to 'weak accept'.

---

> ### Author Rebuttal · Authors · 2024-01-28
>
> Dear Reviewer,
>
> Thank you for providing valuable comments on the experimental results description and evaluation for us. We appreciate your recognition of our work's soundness. Before answering your question, we have responded to your main concerns in the weakness below.
>
> Response to Comment 1: Thanks for your recognition of our methods’ compression performance. RPW requires smaller sizes of auxiliary data than CPDs_PW. We are further improving our methods and making them have less storage cost.
>
> Response to Comment 2: Yes, the search time is primarily impacted by the duration taken to call the CPD lookup function. The fewer binary searches, the faster the search speed will be, which can reduce the search time to a certain extent.
>
> Response to Comment 3: Thank you for your suggestions regarding the presentation and description of experimental performance. We will modify our paper appropriately based on your feedback and address your main concerns as follows.:
>
> a)We show comparison results. Experiments show that RPW has the best preprocessing time on most maps, while CPW has optimal search time on most maps and is far ahead in binary searches. Although our methods may require more preprocessing time, they have absolute advantages in search.
>
> Optimal preprocessing time: BG(120): PW: 42; RPW: 78. DAO(156): PW: 54; RPW: 97; CPW: 5. SC(75): RPW: 75.
>
> Optimal search time: BG: PW: 3; RPW: 50; CPW:67. DAO: PW: 3; RPW: 60; CPW: 93. SC: RPW: 33; CPW: 42.
>
> Optimal binary search: BG: CPW:120. DAO: RPW: 5; CPW: 151. SC: CPW: 75.
>
> b)For your inquiries about the impact of binary searches on improvement levels of 2-3 orders of magnitude: We have listed some data in Table 1 in our previous submitted version.
>
> c)The method used in the online search phase is binary search, which doesn’t rely on any data structure to complete the search in compressed CPD. Therefore, we consider the number of binary searches as a key factor when measuring search cost.
>
> Response to Question : The total query processing time of CPD_PW, CPD_RPW and CPD_CPW are detailed in Table 1.
>
>             Table 1. total query processing time (ns)
>
>       Map    CPDs_PW	CPDs_RPW	CPDs_CPW
>
>       BG	 1.07E+09	6.63E+08	6.48E+08
>
>       DAO    1.18E+10	6.99E+09	6.85E+09
>
>       SC	 3.10E+10	2.29E+10	2.25E+10
>
> We appreciate your review and valuable suggestions. We will carefully evaluate your feedback and make necessary revisions and additions to the final version of the paper.
>
> Sincerely.

---

### Official Review · Reviewer_ehZz · 2023-12-29

**Significance And Importance:** 2
**Soundness:** 3
**Novelty:** 2
**Clarity:** 2
**Overall Evaluation:** 1
**Confidence:** 5

**Weaknesses:**

0: Minor weaknesses requiring some work to be addressed for the paper to be accepted.

**Contributions Of The Paper:**

This paper explores the grid-based pathfinding problem, with a particular emphasis on enhancing the leading algorithm, Compressed Path Database (CPD), through the incorporation of proximity wildcards. The introduction of rectangle and quadrant-based wildcards proves to be effective in reducing storage costs without compromising runtime performance. Nevertheless, the paper currently faces presentation issues that must be addressed significantly to meet the standards required for publication.

**Ethical Considerations:**

(1) Not Applicable: The paper does not have any ethical considerations to address

**Nomination For Best Paper:**

No

**Questions For Authors:**

Please address the  Comments (C) C2, C4, C8, C10 and C12.

**Reproducibility:**

2: Some details are missing, but the paper still appears to be replicable with some effort.

**Strengths Of The Paper:**

1. The concept of the proposed wildcards appears to be interesting.
2. The experimental results show promise.

**Weaknesses Of The Paper:**

Weaknesses:
1. The writing in the paper requires significant improvement.
2. The presentation of experimental results needs enhancement.
3. A more comprehensive comparison with recent developments in Euclidean pathfinding should be considered.

Comments:
1. Abstract,  “In this paper we enhance“, typo “In this paper,”
2. Abstract, “improve search efficiency and reduce the search costs by 2-3 orders of magnitude.” This seems to be inconsistent with experimental results. Do you mean by average, or only for some of maps. Please make the claim clear.
3. Related work, there are many other works that utilise CPD for pathfinding in Euclidean space, road network, traffic management. For completeness, please consider including those.
4. Background - Heuristic move. Equation 2.  Should argmax be argmin?  Also, (s,n) \in E,  E is not defined.
5. Background - Proximity Wildcards. You should explained why proximity wildcards need to be centred on s. Also, please use the toy-dataset to give an example for heuristic move and proximity wildcards.
6. Rectangular ProximityWildcards,  the algorithm 1-2 looks trivial, perhaps it is better to explain them in text or using running example.
7. Coordinates Proximity Wildcards, similarly the algorithm 3-5 are not interesting,  also the figure 5 needs to be better explained to clarify how the proposed algorithm achieve such compression.
8. After reading the two proposed method, my biggest question is why the wildcard must centre/corner on s. Also, why we have to use rectangle or square shaped container, can we associate with other type of geo-entity? such as circle or polygon.
9. Experiments, instead of defining a metric, why not show the original results, e.g., the actual memory cost in MB for CPD. This is important as it can provide a guidance for user when using these suggest techniques. Similar, for others type of measurement.
10. Figure 8 “Runtime (unit: ns). Runtime is the time taken to perform an online search task, and we use it to measure search efficiency”, when you say runtime, is this per first move extraction, or per complete path extraction. Also, the reported performance looks weird to me. To the best of my knowledge so far, I have not aware any existing solver could achieve nano seconds query runtime, including previous CPD works (see https://gppc.search-conference.org/grid). Please investigate carefully.
11. Table 2. Why not simply include TOPPING in Figure 8 ?
12. The authors should consider to compare with more recent works on euclidean pathfinding, such as EPS (Euclidean pathfinding with compressed path databases) and EHL (Efficient object search in game maps). To make it clear, the proposed algorithm doesn’t has to outperform these algorithms, but it would be good to show some comparison so that the reader can refer the performance accordingly. Alternatively, the author should consider to submit the results to GPPC2 (https://gppc.search-conference.org/grid).


---- post rebuttal comments ------

Please make sure you have updated the experiments section accordingly. Also, please carefully proofread the paper and fix these minor comments including C3.

---

> ### Author Rebuttal · Authors · 2024-01-28
>
> Dear Reviewer,
>
> We’d like to express our sincere gratitude for improving our writing and experiment’s completeness with valuable comments. We apologize for any grammatical errors or unclear notations, and will correct them in the next version. Additionally, we are expanding experiments to make our approaches more credible. In response to your major concerns, we address the following.
>
> Response to Comment 2: Only for some of maps. The RPW can reduce optimally by 2 orders of magnitude, while the CPW can reach 3 orders of magnitude. One order of magnitude reduction on average.
>
> Response for Comment 4: Thanks for your care, Equation 2 should be argmin, where E is the set of feasible edges. （s,n)∈E indicates that the path from s to n is feasible.
>
> Response to Comment 8: a) Noted that it is proximity area, not the wildcard centered on s . The wildcard can be used in any qualified location on the map.
>
> b)The reason why we chose to center on s  is that when s is searched and starts to delineate proximity area, the specific direction tendency hasn’t been calculated. An all-round expansion of the proximity area centered on s has high fault tolerance and is more stable. In CPW,  s is the origin of the quadrant.
>
> c)Polygon and circle would be fun to try. However, the more complex the proximity area’s shape often means more preprocessing time and more costs to complete construction. So, it is necessary to comprehensively consider the feasibility.
>
> Response to Comment 10: The runtime in Fig. 8 represents the time required to extract the entire path. We apologize that the scientific notation at the top is too small and confusing your understanding. The time unit is ns, and the scientific notation for Fig. 8 (a) is 1e7, and both Fig. 8 (b) and Fig. 8 (c) are 1e9.
>
> Response to Comment 12: Thank you for your suggestions. Since we can’t obtain exact values from EHL's report, so we only compare with EPS. Our methods usually require more CPD memory but take less time to build than EPS in benchmark BG and DAO. For instance, The average build time of our methods on BG is only one-tenth that of EPS, but CPD memory is 3 times that of EPS. We are expanding experiments and will consider uploading our methods to GPPC2 when they are complete.
>
> We are delighted to receive positive feedback from you, and we appreciate your constructive criticism. We have carefully addressed your major concerns and incorporated your suggestions into the revised version of the paper.
>
> Sincerely.

---

### Official Review · Reviewer_xyVi · 2024-01-19

**Significance And Importance:** 2
**Soundness:** 2
**Novelty:** 2
**Clarity:** 3
**Overall Evaluation:** 1
**Confidence:** 5

**Weaknesses:**

1: Minor weaknesses that are easily fixable.

**Contributions Of The Paper:**

The paper presents two novel ideas for compressing the CPD matrix using new shapes of proximity wildcards. The first idea, called Rectangular Proximity Wildcards (RPW), expands the shape of the proximity wildcards from squares to rectangles. This process increases the number of wildcards in the CPD matrix but comes at the cost of saving an additional value for each node needed to represent width and length of the rectangles. The second idea, called Coordinates Proximity Wildcards (CPW), divides the space in four quadrants with origin in the source node. For each quadrant, this approach computes the largest rectangle that contains expected moves. Similarly to the previous idea, the number of wildcards in the CPD matrix greatly increases at the cost of storing 7 more values for each node w.r.t. Proximity Wildcards.
The increased number of wildcards in the CPD matrix has another advantage other than reducing the dimension of the RLE. In fact, during the path-finding phase, it is possible to automatically infer the first move and avoid the binary search necessary to retrieve it. This should speed-up the path-finding process.
The experimental results analyze the achieved performance on three different sets of game benchmark maps.

**Ethical Considerations:**

(1) Not Applicable: The paper does not have any ethical considerations to address

**Nomination For Best Paper:**

No

**Questions For Authors:**

1) In my understanding, Algorithm 3 returns true for any node within the largest X.n and Y.n independently of the quadrant. For instance, if we consider the first quadrant (i.e. north-east quadrant), the value dx<0 where dx = S.x - T.x means that the target is outside the considered quadrant. The absolute value in line 4 prevents dx from being <0. Can you please clarify this? Also, I think the for loop can be avoided as you always know which quadrant(s) the target belongs to. I hope I didn’t miss anything.
2) What is the first-move metric? As it is defined in Figure 2, the first move array is the list of first optimal moves from the source to any possible target that is then compressed through RLE. If this is the case, I would expect its length to be the same for all considered approaches.
3) Is the runtime reported in Figure 8 in nanoseconds? I would expect that the runtime to compute a path would increase as the maps grow and the CPD gets larger. However, from the plots, I can read that the time to compute a path for the small maps (BGII) is at least 10x the time to compute the path for medium maps. Can you please comment on this?

**Reproducibility:**

3: Authors describe the implementation and domains in sufficient detail.

**Strengths Of The Paper:**

This paper takes a step forward in reducing both the size of the CPD and the time needed to retrieve the first moves and can have an impact in advancing the state-of-the-art for path-planning on grid maps.
The experimental setting increases the previously used benchmark allowing a deeper analysis of the results.
The new contributions of the paper are clearly written and the paper is well organized.

**Weaknesses Of The Paper:**

I think that Algorithm 3 does not match the textual description provided.
Also, the first-move metric is not properly described.
Minors:
There are some typos in the paper
Figure 4 is not clear. I would suggest to divide the representation of each quadrant in a separate grid.

---

> ### Author Rebuttal · Authors · 2024-01-28
>
> Dear Reviewer,
>
> Thank you very much for taking the time and effort to review our paper and provide valuable comments on the presentation and details carefully. Here are our responses to your major concerns :
>
> Question 1: In my understanding, Algorithm 3 returns true for any node within the largest and independently of the quadrant. Can you please clarify this? Also, I think the for loop can be avoided as you always know which quadrant(s) the target belongs to.
>
> Response: We apologize for the inadvertent writing that caused a problem with the presentation of Algorithm 3, and we present it correctly as follows. The For loop determines the proximity area range of CPW, and we consider it cannot be avoided.
> Algorithm 3:
> 1. FOR{n=1 to 4} do
> 2.    X.n  ← rec(s)_n.x
> 3.    Y.n ← rec(s)_n.y
> 4.  ENDFOR
> 5.  Judge t in the nth quadrant of s
> 6.  IF （|s.x-t.x| ≤ X.n \wedge |s.y-t.y| ≤  Y.n）
> 7.    return true
> 8.  ENDIF
> 9.  return false.
>
> Question 2: What is the first-move metric? As it is defined in Figure 2, the first move array is the list of first optimal moves from the source to any possible target that is then compressed through RLE. If this is the case, I would expect its length to be the same for all considered approaches.
>
> Response: Your understanding is correct. The “first move array” refers to the array that stores the first optimal move from the current node to any other feasible goal on the map and is the same for all considered approaches in this paper before compression.
>
> Question 3: Is the runtime reported in Figure 8 in nanoseconds? However, from the plots, I can read that the time to compute a path for the small maps (BGII) is at least 10x the time to compute the path for medium maps. Can you please comment on this?
>
> Response: The time unit is nanoseconds. We apologize that due to poor plot design, the scientific notation at the top of Figure 8 is too small and confusing your understanding of the data. The scientific notation for Figure 8 (a) is 1e7, and both Figure 8 (b) and Figure 8 (c) are 1e9. We have fixed it.
>
> We will carefully consider your feedback and make revisions and supplements in the final version. Thank you for your support and encouragement.
>
> Sincerely.

---

### Meta-Review · Area_Chair_d7GK · 2024-02-03

**Recommendation:** Accept (Oral)
**Confidence:** 3

**Metareview:**

The paper proposes two flexible proximity wildcard methods to improve the compression of CPDs.

Strengths of the paper: New ideas on this relatively not-very-explored of compressed path databases. Significant experimental results.

Weaknesses: The main concern of the reviewers was the quality of the writing. If the paper is accepted we strongly urge the authors to revise their paper.

**Ethical Considerations:**

(1) Not Applicable: The paper does not have any ethical considerations to address